# Quadratus Lumborum Block Reduces Postoperative Opioid Consumption and Decreases Persistent Postoperative Pain Severity in Patients Undergoing Both Open and Laparoscopic Nephrectomies—A Randomized Controlled Trial

**DOI:** 10.3390/jcm10163590

**Published:** 2021-08-15

**Authors:** Michał Borys, Patrycja Szajowska, Mariusz Jednakiewicz, Grzegorz Wita, Tomasz Czarnik, Marcin Mieszkowski, Bułat Tuyakov, Piotr Gałkin, Mansur Rahnama-Hezavah, Mirosław Czuczwar, Paweł Piwowarczyk

**Affiliations:** 1Second Department of Anesthesia and Intensive Care, Medical University of Lublin, 20-059 Lublin, Poland; miroslaw.czuczwar@umlub.pl (M.C.); pawelpiwowarczyk@umlub.pl (P.P.); 2Department of Anesthesia and Intensive Care, Frederic Chopin Clinical Provincial Hospital No. 1, 35-310 Rzeszów, Poland; patrycjaguzek@poczta.onet.pl (P.S.); m.jednakiewicz@szpital.rzeszow.pl (M.J.); sewofluran@interia.pl (G.W.); 3Department of Anesthesiology, Intensive Care and Regional ECMO Center, Institute of Medical Sciences, Opole University, 45-040 Opole, Poland; tczarnik@me.com; 4Department of Anesthesiology and Intensive Care, Regional Specialist Teaching Hospital, 10-719 Olsztyn, Poland; marcinm.mieszkowski@gmail.com (M.M.); bulat_tuyakov@poczta.onet.pl (B.T.); 5Department of Anesthesiology and Intensive Care, School of Medicine, Collegium Medicum, University of Warmia and Mazury, Al. Warszawska 30, 10-082 Olsztyn, Poland; 6Department of Anesthesia and Intensive Care, Jedrzej Sniadecki Hospital, 15-027 Białystok, Poland; piotr.galkin@anestezjolog.com.pl; 7Chair and Department of Dental Surgery, Medical University of Lublin, 20-059 Lublin, Poland; mansur.rahnama@umlub.pl

**Keywords:** nephrectomy, neuropathic pain symptom inventory, patient-controlled analgesia, quadratus lumborum block, persistent postoperative pain

## Abstract

Background: New regional techniques can improve pain management after nephrectomy. Methods: This study was a randomized controlled trial conducted at two teaching hospitals. Patients undergoing elective open and laparoscopic nephrectomy were eligible to participate in the trial. A total of 100 patients were divided into a quadratus lumborum block (QLB) group (50 patients) and a control (CON) group (50 patients). At the end of surgery, but while still under general anesthesia, unilateral QLB with ropivacaine was performed on the side of nephrectomy for patients in the QLB group. The main measured outcome of this study was oxycodone consumption via a patient-controlled anesthesia (PCA) pump during the first 24 h following surgery; other measured outcomes included postoperative pain intensity assessment, patient satisfaction with pain management, and persistent pain evaluation. Results: Patients undergoing QLB needed less oxycodone than those in the CON group (34.5 mg (interquartile range 23 to 40 mg) vs. 47.5 mg (35–50 mg); *p* < 0.001). No difference between the groups was seen in postoperative pain intensity measured on the visual analog scale, except for the evaluation at hour 2, which was in favor of the QLB group (*p* = 0.03). Patients who received QLB were more satisfied with postoperative pain management than the CON group. Persistent postoperative pain was assessed with the Neuropathic Pain Symptom Inventory (NPSI) at months 1, 3, and 6, and was found to be significantly lower in the QLB group at each evaluation (*p* < 0.001). We also analyzed the impact of the surgery type on persistent pain severity, which was significantly lower after laparoscopic procedures than open procedures at months 1, 3, and 6. Conclusions: QLB reduces oxycodone consumption in patients undergoing open and laparoscopic nephrectomy and decreases persistent pain severity months after hospital discharge.

## 1. Introduction

Although a multimodal analgesia approach is recommended following nephrectomy, some individuals still suffer from severe pain after the procedure [1]. Regional anesthesia techniques may improve postoperative pain management and reduce the opioid dosage in the period following renal surgery [2]. Novel blocks have recently been proposed as analgesic options in patients undergoing laparoscopic or open nephrectomy [3,4,5], and such approaches are considered part of multimodal analgesia in the postoperative period [6].

Quadratus lumborum block (QLB) is a relatively new regional anesthesia technique; it was developed by Blanco [7] and has been proposed as an analgesic alternative for use after many types of surgical procedures [8,9,10]. Several approaches to QLB have been reported [11], but only a few randomized controlled trials (RCT) [4,5,12,13] have evaluated the use of QLB for nephrectomies, all of which concerned laparoscopic procedures. Furthermore, there were differences in the type of block, injection site, and postoperative pain management, and, to our knowledge, none of the RCTs considered QLB following open nephrectomies.

Our study aimed to evaluate the impact of QLB on postoperative oxycodone consumption in patients undergoing laparoscopic and open nephrectomies. The secondary measures included postoperative pain intensity assessment, patient satisfaction with pain management, and persistent pain evaluation. Our primary hypothesis was that a 25% reduction in oxycodone use would be observed in the QLB group.

## 2. Materials and Methods

### 2.1. Ethical Considerations

The RCT was conducted in the urology departments of two teaching hospitals. The Institutional Review Board of the Medical University of Lublin, Poland approved the study protocol (permit number KE-0254/328/2017), which was registered with ClinicalTrials.gov (NCT03529201) before patient recruitment. Written informed consent was obtained from each patient, and the study was conducted in accordance with the principles for medical research involving human subjects of the Declaration of Helsinki.

### 2.2. Patient Selection

Adult patients younger than 80 years who were scheduled for nephrectomy procedures (due to cancer) could volunteer to participate in the study. Patients with known coagulopathy, allergies to the studied drugs, psychiatric disorders requiring antidepressant usage, or addiction to alcohol or other recreational drugs were excluded. Before recruitment, the investigators assessed the patients’ ability to handle a patient-controlled analgesia (PCA) pump and to understand the visual analog scale (VAS) and the Neuropathic Pain Symptom Inventory (NPSI).

### 2.3. Interventions

General anesthesia was induced and maintained by an anesthesiologist who was unaware of the patient’s group allocation. To induce general anesthesia, each patient received fentanyl (1–3 µg/kg Fentanyl WZF, Warszawskie Zakłady Farmaceutyczne Polfa S.A., Warsaw, Poland), propofol (1–2 mg/kg Propofol 1% Fresenius, Fresenius Kabi Deutschland GmbH, Bad Homburg, Germany), and rocuronium (0.6–1.0 mg/kg Esmeron, N.V. Organon, Jersey City, NJ, USA). Anesthesia was maintained with 1.5–2.0% sevoflurane (Sevorane, AbbVie, Chicago, IL, USA) in a mixture of oxygen and air, and additional doses of fentanyl and rocuronium were administered. Standard monitoring during the procedure included blood oxygen saturation (SpO_2_), a three-lead electrocardiogram (ECG), non-invasive blood pressure (NIBP), capnography, oxygen, and volatile anesthetic concentrations. The type of surgery—open or laparoscopic nephrectomy—was chosen at the surgeon’s discretion.

At the end of the procedure, an opaque envelope with the chosen randomization was opened. Half the patients were assigned to the control (CON) group and half to the QLB group. Computer-generated randomization was prepared by an investigator who was not involved in the procedure or further assessment of the participants. Both the primary investigator and the statistician were blinded to data collection throughout the study.

Patients in the CON group emerged from anesthesia and were transferred to the post-anesthesia care unit (PACU). In the other group, QLB was performed before the end of anesthesia, under ultrasound guidance, and with the patient in the lateral position. Only one person in each center performed the QLBs. The anesthesiologist used a 10 cm Ultraplex 360 needle (B Braun, Melsungen, Germany) to perform a QLB type II (on the posterolateral surface of the QL muscle). The needle was introduced in the medial to posterolateral direction. Infiltration with 0.9% saline and the needle’s position were confirmed, and 2 mL/kg 0.375% ropivacaine (Ropimol, Motleni, Firenze, Italy) was then deposited (up to 40 mL) only on the side of the surgery. Patients were not aware of the group to which they had been allocated.

### 2.4. Postoperative Care and Pain Management

In the PACU, oxygen was supplied and the patient’s vital signs, including respiratory rate, plethysmography, oxygen saturation, heart rate, and blood pressure, were monitored. The standard postoperative pain management in both groups included oxycodone and non-opioid drugs. The investigator began pain treatment with oxycodone administered intravenously (i.v.) via a PCA pump, 1 mg/mL with a 5-minute lockout. The attending nurse could add 5 mg of oxycodone in the case of severe pain exceeding 40 mm on the VAS. The other painkillers comprised i.v. paracetamol, 1 g every 6 h, and i.v. ketoprofen, 100 mg twice daily. The patient also received 4 mg of ondansetron, twice daily, as prophylaxis for nausea and vomiting. The nurse, who was unaware of the participants’ group allocations, reminded the patients about the pain severity assessment.

### 2.5. Outcomes

The total patient consumption of oxycodone via PCA during the first 24 postoperative hours was the primary measure of the study. The other outcome measures included pain severity on the VAS at hours 2, 4, 8, 12, and 24; rescue doses of oxycodone administered by nurses in the PACU; patient satisfaction with pain management; and persistent pain evaluation. Satisfaction with pain treatment was assessed on a five-point Likert scale on which the patient and the investigator could describe the pain management as very poor (1), poor (2), moderate (3), good (4), or excellent (5). The presence and intensity of persistent postoperative pain were evaluated during a phone interview at months 1, 3, and 6 using the NPSI [14], which was also used in our previous studies on persistent postoperative pain [15,16].

### 2.6. Statistical Analysis

Continuous data were compared using the Mann–Whitney U test or the Kruskal–Wallis test by ranks (for more than two comparisons). Medians (interquartile ranges) are used to present these data. Categorical variables were compared using Fisher’s exact test. Logistic regression was used to reveal the parameters that affect chronic pain presentation. The odds ratio (OR) was used to describe the predictors that were included in the model. The receiver operating characteristic (ROC) curve was calculated for the best model. All analyses were performed using Statistica 13.1 software (Stat Soft. Inc., Tulsa, OK, USA), which was also used to generate the random group assignments.

### 2.7. Power Analysis

A sample size analysis was performed for the primary outcome of the study. In a pilot study, we recruited 20 patients (10 per group) and found a mean oxycodone consumption of 42 mg in the CON group and 30 mg in the QLB group, with an SD of 18. The calculated sample size for a significance level of 0.05 and power of 0.9 that would detect a hypothesized 25% reduction in oxycodone use was 49 participants per group; we therefore recruited 105 patients with a 1:1 randomization ratio.

## 3. Results

The study was conducted from May 2018 to February 2019. Figure 1 presents the study flowchart. Patient demographics are presented in Table 1. We found no significant differences between the QLB and CON groups in terms of demographics or the type or duration of surgery or anesthesia, and postoperative complications. 

### 3.1. Oxycodone Consumption with PCA

Figure 2 presents the primary outcome of this study—24-hour oxycodone consumption with a PCA pump. Patients in the QLB group required less oxycodone than those in the CON group (34.5 mg (23–40 mg) vs. 47.5 mg (35–50 mg); *p* < 0.001).

### 3.2. Acute Postoperative Pain and Patient Satisfaction

Patients in the QLB group had less severe pain than those in the CON group at hour 2. Table 2 shows the pain intensity evaluated using the VAS. The QLB group also required fewer rescue doses of oxycodone than the CON group (0 (0–5) vs. 5 (0–5); *p* = 0.028), and patients in the QLB group were more satisfied with the postoperative pain management (Table 3).

### 3.3. Presence and Severity of Persistent Postoperative Pain

Patients in the QLB group had significantly lower pain severity, as measured with the NPSI, than the controls at months 1, 3, and 6 (Table 4). However, no difference was found in the number of patients with signs of chronic pain (more than 0 on the NPSI). We divided patients according to the surgery type—49 laparoscopic and 51 open nephrectomies across the study groups—and analyzed the risk of persistent postoperative pain on that basis. As shown in Table 5, open nephrectomies predisposed patients to higher persistent pain intensity than laparoscopic surgeries, but instances of chronic pain (the number of patients with any signs of persistent pain) did not differ either between the QLB and CON groups nor between the open and laparoscopic surgeries.

The combined results for persistent pain severity, including both type of surgery and study group, are shown in Table 6. Pairwise comparisons between the four groups of patients at months 3 and 6 are graphically presented in Figure 3.

Significant differences are represented by arrows pointed toward the lower pain severity at months 3 (green arrows) and 6 (red arrows). Pairwise comparisons were made with the Mann–Whitney U test using a significance level of 0.008 after Bonferroni correction. CON: control group; QLB: quadratus lumborum block group.

### 3.4. Persistent Pain Predictors

Three factors were associated with the incidence of persistent pain after nephrectomy at month 6. The severity of persistent pain at month 3 increased the risk of chronic pain prevalence at month 6 (OR 1.28 (1.13, 1.45)). Conversely, a lower amount of oxycodone administered via PCA had a reduced risk of persistent pain after 6 months (OR 0.90 (0.85, 0.96)). Moreover, patients after laparoscopic nephrectomy who received QLB had a lower chance of chronic pain (0.11 (0.02, 0.51])). The area under the ROC curve for this model was 0.91 (Figure 4).

## 4. Discussion

The primary hypothesis of our study—A 25% reduction in PCA-administered oxycodone in the QLB group—was confirmed by the results (Figure 2). Furthermore, the CON group required more rescue doses of oxycodone than the QLB group, who were also more satisfied with their postoperative pain treatment (Table 3), although statistically significant alleviation of pain severity in the QLB group was only seen at hour 2 (Table 2). Persistent pain severity measured using the NPSI was lower in the QLB group than in the CON group at months 1, 3, and 6, but the numbers of patients with any signs of chronic pain (NPSI > 0) did not differ between the groups.

Further analysis showed that patients undergoing open nephrectomies had more severe chronic pain at months 1, 3, and 6 than those undergoing laparoscopic surgery (Table 4). In our study, two factors had an impact on chronic pain severity, with both QLB and laparoscopic surgery lowering pain intensity months after surgery (Table 6). At month 6 (Figure 3), chronic pain severity was lower in the QLB laparoscopic nephrectomy group than in the CON group (regardless of surgery type), but no difference was found between laparoscopic and open nephrectomies in the QLB group.

Our study had some methodological differences from previous RCTs examining QLB use in patients after nephrectomy. First, both open and laparoscopic procedures were studied; in previous RCTs, only laparoscopic nephrectomies were investigated [4,5,12,13]. Second, in our study, the posterolateral QLB was performed postoperatively for both open and laparoscopic nephrectomies. In most other RCTs on this topic, the QLB was performed before laparoscopic nephrectomy [4,5,13], although Aditianingsih et al. also performed the QLB postoperatively [12]. Third, in Aditianingsih et al. and in a study by Dam et al., the QLB was done bilaterally, while Zhu et al. and Kwak et al. performed the QLB only on the side of the surgery, as in our study. Fourth, Kwak et al. and Aditianingsih et al. performed medial QLB, Dam et al. and Zhu et al. performed transmuscular QLB, but posterior QLB was used in our study.

Similar to our results, in all RCTs in which QLB was compared with a control, the total consumption of opioids administered via PCA was significantly lower at hour 24 following the surgery [4,5,13]; notably, Aditianingsih et al. compared QLB with epidural analgesia [12] and found no difference in total opioid consumption between the groups. All the RCTs on this topic identified, at most, moderate pain relief following QLB compared to controls.

Few researchers have investigated the problem of persistent pain in patients following nephrectomy. Alper and Yüksel compared 27 laparoscopic nephrectomy patients with 25 open surgery patients [1] and found no difference between the types of nephrectomies in terms of either acute postoperative pain or chronic pain measured at month 6, with only two patients reporting chronic pain. In contrast, as shown in Table 5, most patients in our study had some signs of chronic pain at month 6.

Furthermore, more severe pain was found after open nephrectomy than laparoscopic nephrectomy in our RCT 6 months after surgery, but only in the CON group. The difference between our results and those of Alper and Yüksel could be a result of the NPSI in our trial allowing more detailed identification of persistent pain. However, Owen et al. also found chronic pain in 33% of kidney donors following open surgery [17], and the donors with chronic pain—especially neuropathic in nature—experienced a meaningful loss of quality-adjusted life years (QALY).

Our study presented a significant reduction in persistent pain severity in patients following the plain block, however, the QLB was performed postoperatively. It is possible that chronic pain severity would have been even lower in the QLB group if the block had been done pre-emptively. However, the block was performed under general anesthesia. Thus, patients had reduced nociception due to opioids and anesthetics. 

Our study has some limitations. Patients undergoing both laparoscopic and open nephrectomies were included, and our population was thus more heterogeneous than in previous RCTs. Patients in the CON group also did not receive a sham block with saline. Although a significant difference in persistent pain severity between the QLB and the CON groups was found in our study, its impact on patient recovery and quality of life was not assessed. To conclude, QLB reduces opioid consumption and alleviates persistent pain severity in patients undergoing laparoscopic and open nephrectomies.

## Figures and Tables

**Figure 1 jcm-10-03590-f001:**
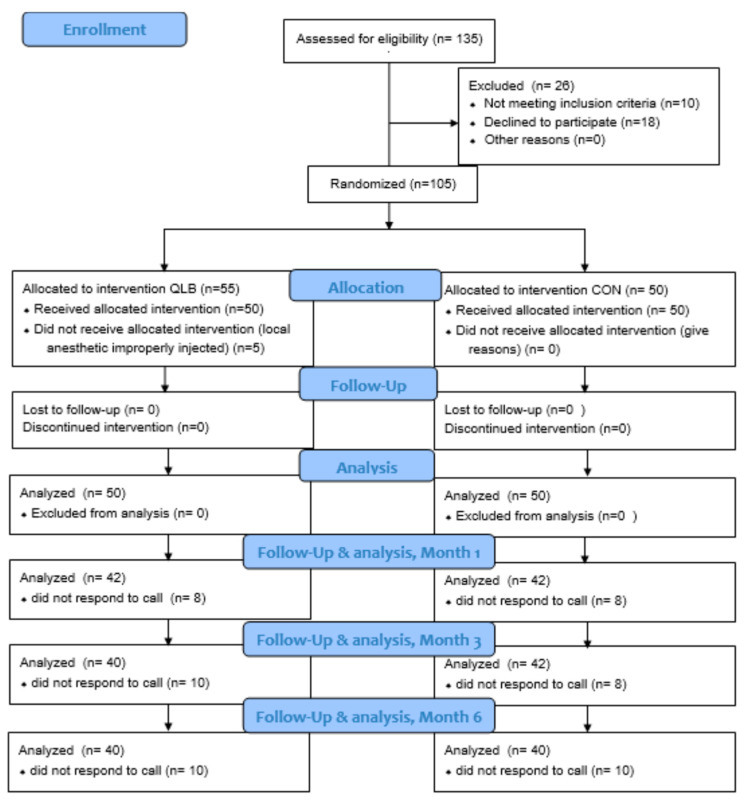
Study flowchart; CON: control group; QLB: quadratus lumborum block group.

**Figure 2 jcm-10-03590-f002:**
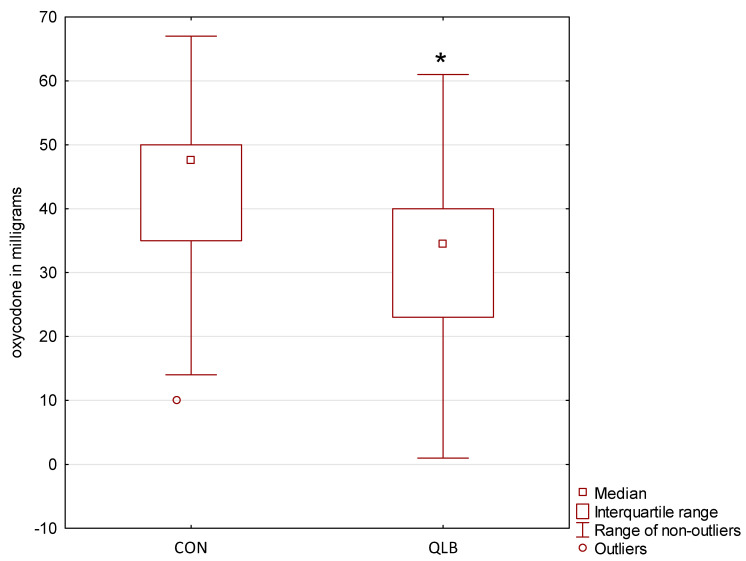
Consumption of oxycodone via patient-controlled analgesia pump during the first 24 postoperative hours; CON: control group; QLB: quadratus lumborum block group; * statistically significant.

**Figure 3 jcm-10-03590-f003:**
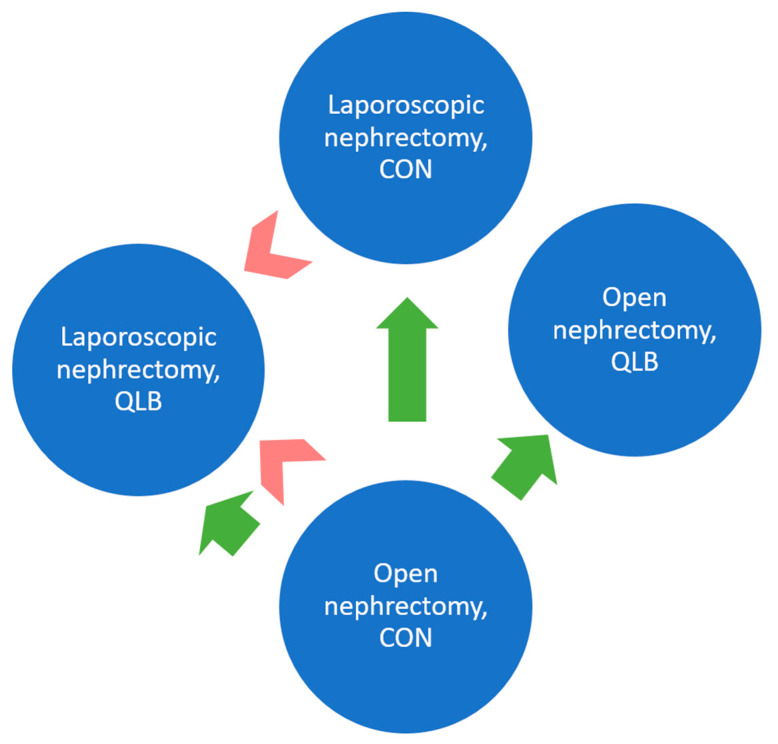
Persistent pain severity measured with the NPSI at months 3 and 6.

**Figure 4 jcm-10-03590-f004:**
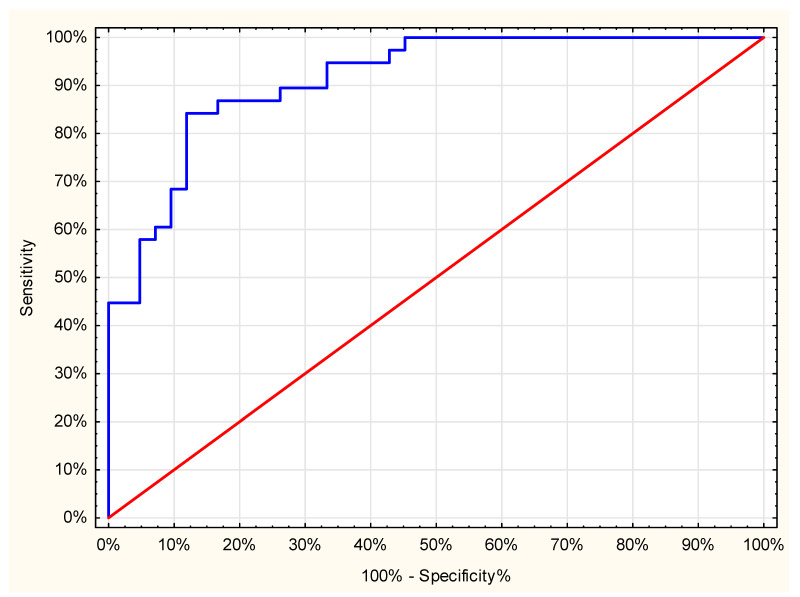
Receiver operating characteristic curve; The receiver operating characteristic curve of the persistent pain model was calculated with logistic regression.

**Table 1 jcm-10-03590-t001:** Patient demographics and surgery times.

Feature	QLB (n = 50)	CON (n = 50)	*p*-Value
Age (yr)	61 (49–71)	62 (57–68)	0.69
Female	17 (34%)	15 (30)	0.83
Weight (kg)	80 (72–88)	80 (74–90)	0.28
Height (cm)	171 (164–176)	169 (164–176)	0.69
BMI	27.1 (24.2–30.5)	28.0 (26.1–30.8)	0.19
Laparoscopic surgery	27 (54%)	22 (44%)	0.42
Time of surgery (min)	90 (75–100)	90 (80–100)	0.62
Time of anesthesia (min)	120 (100–135)	115 (100–125)	0.09
Intraoperative fentanyl consumption (mcg)	200 (200–250)	200 (200–250)	0.53
Postoperative nausea/vomiting	4/1	7/2	0.52/1.0

Data are shown as medians (interquartile ranges) or numbers (percentages). Probability was calculated with the Mann–Whitney U test or Fisher’s exact test. BMI: body mass index; CON: control group; QLB: quadratus lumborum block group.

**Table 2 jcm-10-03590-t002:** Pain intensity after nephrectomy measured with the visual analog scale.

Hours Post-Surgery	QLB (mm)	CON (mm)	*p*-Value
2	40 (25–60)	50 (32–60)	0.03
4	40 (20–50)	40 (30–50)	0.06
8	31 (20–43)	40 (24–50)	0.21
12	30 (20–40)	30 (20–40)	0.65
24	24 (12–30)	25 (20–30)	0.50

Data are shown as medians (interquartile ranges). Probability was calculated with the Mann–Whitney U test. CON: control group; QLB: quadratus lumborum block group.

**Table 3 jcm-10-03590-t003:** Patient satisfaction with postoperative pain treatment.

Feature	QLB	CON	*p*-Value
Assessed by patient	4 (4–5)	4 (3–4)	<0.001
Assessed by researcher	4 (4–5)	4 (3–4)	<0.001

Data are shown as medians (interquartile ranges). Probability was calculated with the Mann–Whitney U test. CON: control group; QLB: quadratus lumborum block group.

**Table 4 jcm-10-03590-t004:** Patients with chronic postoperative pain and pain severity measured with the NPSI by study group.

Months after Discharge	QLB	CON	*p*-Value
Median (interquartile range) persistent pain severity			
1	25 (19–35)	43 (34–48)	<0.001
3	15 (13–22)	29 (17–32)	<0.001
6	7 (4–11)	15 (7–21)	<0.001
Number (%) of patients with persistent pain			
1	41 (98)	40 (95)	1
3	38 (95)	40 (95)	1
6	34 (85)	38 (95)	0.26

Probability was calculated with the Mann–Whitney U test for pain severity and Fisher’s exact test for persistent pain incidence. CON: control group; NPSI: Neuropathic Pain Symptom Inventory; QLB: quadratus lumborum block group.

**Table 5 jcm-10-03590-t005:** Patients with chronic postoperative pain and pain severity measured with the NPSI by surgery type.

Months after Discharge	Laparoscopic Surgery	Open Surgery	*p*-Value
Median (interquartile range) persistent pain severity			
1	25 (14–38)	42 (26–47)	<0.001
3	17 (12–23)	24 (16–32)	<0.01
6	7 (4–11)	14 (7–20)	<0.01
Number (%) of patients with persistent pain			
1	38 (93)	43 (100)	0.11
3	37 (90)	41 (100)	0.12
6	34 (85)	38 (95)	0.26

Probability was calculated with the Mann–Whitney U test for pain severity and Fisher’s exact test for persistent pain incidence. NPSI: Neuropathic Pain Symptom Inventory.

**Table 6 jcm-10-03590-t006:** Patients with chronic postoperative pain and pain severity measured with the NPSI by study group and surgery type.

Months after Discharge	QLB		CON		*p*-Value
	Laparoscopic Surgery	Open Surgery	Laparoscopic Surgery	Open Surgery	
Median (interquartile range) persistent pain severity					
1	22 (14–28)	35 (25–39)	39 (10–43)	47 (43–49)	<0.001
3	14 (10–20)	18 (14–23)	19 (12–30)	32 (23–35)	<0.001
6	5 (3–9)	8 (5–14)	11 (6–16)	18 (9–23)	<0.001
Number (%) of patients with persistent pain					
1	22 (96)	19 (100)	16 (89)	24 (100)	0.16
3	21 (91)	17 (100)	16 (89)	24 (100)	0.22
6	19 (83)	15 (88)	15 (88)	23 (100)	0.19

Probability was calculated with the Mann–Whitney U test for pain severity and Fisher’s exact test for persistent pain incidence. CON: control group; NPSI: Neuropathic Pain Symptom Inventory; QLB: quadratus lumborum block group.

## Data Availability

Data are available from the first author after a reasonable request.

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
