# Peer review of "Quadratus Lumborum Block Reduces Postoperative Opioid Consumption and Decreases Persistent Postoperative Pain Severity in Patients Undergoing Both Open and Laparoscopic Nephrectomies—A Randomized Controlled Trial"

_jcm, 2021, doi:10.3390/jcm10163590_

Round 1
Reviewer 1 Report
The authors addressed all the comments.
Thank you
Author Response
Thank you for your comments.
Reviewer 2 Report
The authors have now provided a ROC curve and this is okay for me
Author Response
Thank you for your comments.
This manuscript is a resubmission of an earlier submission. The following is a list of the peer review reports and author responses from that submission.
Round 1
Reviewer 1 Report
The authors present an interesting work analyzing the role of QLB in patients undergoing nephrectomy.
In my opinion there is an important limitation due to the inclusion , in the same analysis, of patients undergoing laparoscopic and open procedures.
It would be interesting to develop sub-group analyses in order to find possible differences related to the surgical technique ( use of opioids, pain intensity..).
I have some questions:
- did patients in the control group receive a bolus of oxycodone before the end of anesethesia?
- Did you measure the use of oxycodone according to the different post-operative day?
- Did you measure the incidence of side effects such as PONV?
- Did you evaluate if the major amount of oxycodone had a negative impact on patients'postoperative course (length of hospital stay, restart of oral feeding)
The data on chronic postoperative pain are very interesting.
Thank you
Author Response
Point 1. The authors present an interesting work analyzing the role of QLB in patients undergoing nephrectomy.
In my opinion there is an important limitation due to the inclusion , in the same analysis, of patients undergoing laparoscopic and open procedures.
Our response:
Thank you for your comments. We hope that they will improve our manuscript. Yes, this is a limitation of our study, however, we mentioned it in the limitations section.
Point 2. It would be interesting to develop sub-group analyses in order to find possible differences related to the surgical technique ( use of opioids, pain intensity..).
Our response: We checked these differences related to the surgical technique. No differences were found between both approaches. Because many data and parameters were included in the manuscript, to avoid confusion in readers, we did not decide to mention them in the text.
Point 3. did patients in the control group receive a bolus of oxycodone before the end of anesethesia?
Our response: Both groups did not receive bolus of oxycodone before the end of anesthesia.
Point 3. Did you measure the use of oxycodone according to the different post-operative day?
Our response: Oxycodone was only measured during the first 24 postoperative hours.
Point 4. Did you measure the incidence of side effects such as PONV?
Our response: Yes, it was measured. We added the piece of information concerning postoperative complications to the text. Moreover, we added the incidence of PONV to Table 1.
Point 5. Did you evaluate if the major amount of oxycodone had a negative impact on patients'postoperative course (length of hospital stay, restart of oral feeding)
Our response: No difference was noticed in the length of stay. However, we did not evaluate the impact of oxycodone consumption on the restart of oral feeding.
Point 6. The data on chronic postoperative pain are very interesting.
Our response: Thank you for your opinion.
Reviewer 2 Report
Abstract
please report number of patients in each group
please report of total amount of fentanyl for each group
did patients had any reinjection ? this should be reported whrther the response is yes or no as the significance of the difference in the first 2 hours could be in relation to the initial amount of fentanyl injected
It appears to me the sample size selection was for the purpose of primary objective therefore there is probably not enough data to elaborate adequate conclusion about the secondary outcomes which are persistant and chronic pain
withb this regard for example a median difference of 17 points with a wide interquartile range might not be clinically relevant , by the why this statistical confirmation as these are repeated measuremment and ANOVA should be used but since because of lack of normality ANOVA with repeated measurement cannot be used and Mann Whitney was used , I think these chronic and pesistent pain results need extensive statistical analysis
Author Response
Thank you for your comments.
Point 1. Abstract please report number of patients in each group
Our response: we added a number of patients to the abstract
Point 2. please report of total amount of fentanyl for each group
Our response: We added the intraoperative fentanyl consumption to Table 1. No difference was noted between both groups.
Point 3. did patients had any reinjection ? this should be reported whrther the response is yes or no as the significance of the difference in the first 2 hours could be in relation to the initial amount of fentanyl injected
Our response: Patients received 1-2 mcg per kg for the induction of GA. Because some surgeries lasted over 2 hours, patients received extra injections of fentanyl during the procedure. We monitored the total amount of fentanyl, but not the spread of additional injections in time. Could it have an impact on pain severity? Anything is possible.
Point 4. It appears to me the sample size selection was for the purpose of primary objective therefore there is probably not enough data to elaborate adequate conclusion about the secondary outcomes which are persistant and chronic pain
Our response: Yes, the power analysis refers to the primary objective of this study. However, we believe that data concerning might be interesting for the readers. It appears that this issue was not researched extensively in such a group of patients.
Point 5. withb this regard for example a median difference of 17 points with a wide interquartile range might not be clinically relevant , by the why this statistical confirmation as these are repeated measuremment and ANOVA should be used but since because of lack of normality ANOVA with repeated measurement cannot be used and Mann Whitney was used , I think these chronic and pesistent pain results need extensive statistical analysis
Our response: We are aware that such a difference might not be relevant. Thus we stated in the limitations that “recovery and quality of life were not assessed”. These evaluations could show the impact of persistent pain on life. ANOVA with repeated measures would have been statistics of choice if these data had been normally distributed. As suggested by the reviewer, we analyzed the data using other tests. The results of logistic regression are added to the manuscript.
Reviewer 3 Report
Throughout my first reading of the study, I was thinking that there was a flaw in the study, since there was no "sham" injection of saline, instead of the local anesthesia. The researchers elected to either perform the injection with an anesthetic, or return the patient to the hospital bed without injection. I felt this presented the risk that the patients (or family or other caregivers) could easily determine if the patient had been in the study or not, both immediately post-op and even for days afterward. It might be easy to see if there was evidence of needles puncturing the skin or not. I was glad to see the authors address this concern in their discussion. I believe it would have been a stronger study if the same treatment had been administered to both study and control subjects, with the personnel performing the ultrasound guided blocks blinded as to whether they were injecting local anesthetic or saline. Overall however, it is an interesting study that suggests administration of an ultrasound guided quadratus lumborum block contributes to a significant reduction in immediate post-operative pain following nephrectomy. The study also appears to clearly demonstrate that while the block reduces acute postoperative pain in both open and laparoscopic procedures, there should be no claims made that this procedure makes a significant difference in reducing chronic pain after either the open or laparoscopic approach to performing a nephrectomy.
Author Response
Throughout my first reading of the study, I was thinking that there was a flaw in the study, since there was no "sham" injection of saline, instead of the local anesthesia. The researchers elected to either perform the injection with an anesthetic, or return the patient to the hospital bed without injection. I felt this presented the risk that the patients (or family or other caregivers) could easily determine if the patient had been in the study or not, both immediately post-op and even for days afterward. It might be easy to see if there was evidence of needles puncturing the skin or not. I was glad to see the authors address this concern in their discussion. I believe it would have been a stronger study if the same treatment had been administered to both study and control subjects, with the personnel performing the ultrasound guided blocks blinded as to whether they were injecting local anesthetic or saline. Overall however, it is an interesting study that suggests administration of an ultrasound guided quadratus lumborum block contributes to a significant reduction in immediate post-operative pain following nephrectomy. The study also appears to clearly demonstrate that while the block reduces acute postoperative pain in both open and laparoscopic procedures, there should be no claims made that this procedure makes a significant difference in reducing chronic pain after either the open or laparoscopic approach to performing a nephrectomy.
Our response: Thank you for your comments. We are aware that our study had some shortcomings. Most of them were described in the limitations. We decided not to perform a sham block because our patients were still under GA. We believe that it did not affect the patient’s blinding in this case.
Round 2
Reviewer 2 Report
The authors have imroved the manuscript
there is a question of ROC curve but I don't see any curve please explain or add the curve
yours
Author Response
The authors have imroved the manuscript
there is a question of ROC curve but I don't see any curve please explain or add the curve
yours
Our response:
ROC curve was added to the manuscript.